# Adaptive fuzzy flow rate control considering multifractal traffic modeling and 5G communications

**Alisson Assis Cardoso**◉*, **Flávio Henrique Teles Vieira**◉

School of Mechanical, Electrical and Computer Engineering, Federal University of Goiás, Goiânia, Goiás, Brazil

◉ These authors contributed equally to this work.
* alsnac@gmail.com

**Data Availability Statement:** All relevant data are within the manuscript and its Supporting Information files.

**Funding:** Alisson Assis Cardoso was financed by the Coordenação de Aperfeiçoamento de Pessoal

## Abstract

In this paper, we propose a predictive Generalized OBF (Orthonormal Basis Functions)-Fuzzy flow control scheme for the 5G downlink by deriving an expression for the optimal control rate of the traffic sources considering minimization of data delay and a minimum traffic rate to the users. The adaptive GOBF-Fuzzy model is applied to predict queueing behavior in initial 5G systems. To this end, we propose to obtain orthonormal basis functions related to the real traffic flows via multifractal modeling, inserting these functions into the fuzzy model trained with the LMS (Least Mean Square) adaptive algorithm. Simulations of a F-OFDM (Filtered Orthogonal Frequency Division Multiplexing) based 5G Downlink are carried out to validate the proposed flow control algorithm. Comparisons with other predictive control schemes in the literature prove the efficiency of the adaptive GOBF-fuzzy based control in enhancing the performance of the system downlink as well as guaranteeing some QoS (Quality of Service) parameters.

## 1 Introduction

Future 5G systems will provide high data rates and low latency through optimized packet radio access and flexible bandwidth. Such features can be attained due to some techniques regarding data transmissions, such as F-OFDM and W-OFDM (Filtered and Window Orthogonal Frequency Division Multiplexing) [1]. In this sense, network traffic control algorithms will become even more essentials for guaranteeing these features to users.

Data addressed to mobile users in the 5G system is stored in queues at the base station (5G eNodeB—Evolved Node B) until the transmission rates are provided. These transmission rates are influenced by the quality of the user channel, so users with better channel conditions tend to obtain higher transmission rates. If these transmission rates are not sufficient for attaining the user demands, queue congestion can occur, causing data loss. For this reason, flow rate control algorithms can be applied to the arriving network traffic flows at the base station in order to provide more adequate service to mobile users. Flow rate control algorithms when

de Nível Superior - Brazil (CAPES) - Finance Code 001 (https://www.capes.gov.br/). The funders had no role in study design, data collection and analysis, decision to publish, or preparation of the manuscript.

**Competing interests:** The authors have declared that no competing interests exist.

applied to 5G systems allow that smaller queue sizes at eNodeB be obtained [2]. Smaller queue sizes induce the system to attain lower loss rates and shorter waiting times to users.

Network traffic control can be enhanced when considering a precise traffic modeling, such as that provided by fuzzy approaches. Fuzzy modeling has been widely applied to many researches since it presents certain advantages over linear models, for example, in the description of unknown real processes with nonlinear and time-varying characteristics such as network traffic [3]. In [4], the authors propose an hybrid technique combining the Type-2 Fuzzy C-Means and Artificial Neural Network to improve the prediction of highway speed traffic flow compared to classical methods in the literature. In [5], it is proposed a prediction method that combines denoising schemes and support vector machine, outperforming models that do not consider the denoising strategy.

In the last decades, several studies have shown the importance of traffic process analysis using the wavelet transform due to its multiscale representation [5–8]. One of the applications of wavelet transform is in network traffic modeling in order to describe behaviors such as long-range dependence and burst incidences at different time scales [9, 10]. These characteristics may degrade network performance in relation to Gaussian and short-range dependence traffic flows [8, 11]. Multifractal models precisely describe traffic flows in small scales (*ms* or smaller), being adequate for the initial 5G systems, whose scheduling time is of the order of $1ms$ [10, 12].

The main multifractal models are based on multiplicative cascades, which are structures where an interval is randomly divided by multipliers, conserving the interval mass [8]. Thus, at the end of the division process, a correlated sequence is obtained, representing the network traffic samples. As examples of wavelet domain based multifractal models, we can cite: The Lognormal Beta [13] model and the MWM (Multifractal Wavelet Model) [8]. The MWM model consists of a multiplicative cascade in the Haar wavelet domain [14], where multiplicative cascade multipliers are computed based on the signal energy decay. Although the MWM model being suitable for modeling network traffic, it requires the application of the wavelet transform to the whole traffic trace or to all samples in a time window that is intending to apply the model. In other words, in the original formulation of the MWM, its parameters are not updated at each time instant that a traffic sample is provided. This motivated us to propose an adaptive wavelet based multifractal modeling approach that is precise even being adequate for real time applications.

In order to achieve high utilization of resources in communication networks and for better decision making, traffic prediction can be used and must be as accurate as possible. Fuzzy modeling is capable of precisely representing a nonlinear complex process such as network traffic traces through the combination of linear local models [3]. In [15], the authors highlight the importance and principles of fuzzy logic applications in the area of channel estimation, channel equalization, handover management and QoS (Quality of Service) management. Moreover, adaptive prediction algorithms are more appropriate for real time multimedia applications than on-batch prediction algorithms due to on-line processing capability and varying nature of network traffic. Taking these into account, we also address the development of an adaptive fuzzy prediction algorithm that incorporates a wavelet domain modeling of network traffic.

In [16], the authors propose a scheduling algorithm with flow rate control for LTE downlink systems taking into account the size of each user queue. Thus, users with greater queue sizes will have higher priority compared to others. Also, in [16], the authors propose to use flow rate control algorithms to control network traffic that is not sensitive to delay (best effort). The results presented by the authors show that control algorithms can provide a significant improvement in the waiting time in the queues.

There are various proposals of control schemes in the literature that are dedicated to network protocols, such as that presented in [17], that is based on the flow control mechanisms of the Transmission Control Protocol/Internet Protocol (TCP/IP). Among the proposals for flow rate control that do not depend on specific network mechanisms, we can mention the Proportional Control method [17, 18]. Such methods can be used to control real-time applications and are also effective for other control problems.

The authors in [19] propose the use of the Kalman filter in order to predict the end-to-end delay in networks. By estimating the delay, an analysis of the buffer occupancy is carried out to send the information of the intensity of the flow rate to the transmitter. In this way, the control scheme can regulate the flow rate based on this analysis. Thus, the transmitter user makes a balance between the estimated rate that optimizes the queuing delays and a rate that minimizes the loss rate.

Some works in the literature aim to control the flow rate in 5G systems with the use of Software-Defined Networking (SDN) as is the case of [20]. In [20], the authors propose to control network traffic flows considering the optimization of some parameters, such as energy consumption of the users' equipment. In [21], the authors propose a scheduling algorithm with flow rate control for LTE downlink systems taking into account the size of each user queue. The results presented by the authors show that control algorithms can provide a significant improvement in the waiting time in the queues. The authors in [19] propose the use of the Kalman filter in order to predict the end-to-end delay in networks. The control scheme can regulate the flow rate making a balance between the estimated rate that optimizes the queuing delays and a rate that minimizes the loss rate. Although, there is an improvement in the quality of service parameters shown in [20], the proposal overuses the control information exchange between the central control and the local control in the user equipment. Also, once the simulated annealing is used, a high computational effort is required, becoming the flow rate control unfeasible to real-time applications. Aiming to reduce the computational effort, in [22, 23] the authors present a flow control algorithm with low computational complexity for wireless networks. To this end, they propose the use of the Linear Quadratic Gaussian (LQG) to control the network flow rate. Although the computational effort is reduced, the need to perform a pre-tuning parameter algorithm and the linear modeling of the flow can degrade the performance for low-scale aggregated and non-linear network traffic whose variation is high, as in 5G systems.

In this work, we propose a fuzzy based algorithm to control the traffic flow rates in order to minimize a cost function given in terms of the prediction of the queueing behavior in the buffer. Differently from the previous works, the aim of this work is to provide a more opportunistic control with the use of the fuzzy logic, providing adequate traffic flow rates according to the buffer size in order to maintain it below a desired level. Another important factor, not considered by the mentioned works, is that we also consider in the cost function the minimum rate required for each user. In addition, we verify that the use of adaptive multifractal modeling together with the orthonormal basis functions allows us to enhance real-time prediction of the queueing buffer behavior. To this end, firstly in the next section, we propose an algorithm to adaptively estimate the parameters of the Lognormal Beta Multifractal model [10] that precisely describe traffic flows in small scales (adequate for 5G communications).

In summary, the main contributions of the present paper are:

1. Equations to adaptively estimate the moment factor and the scaling function of network traffic flows;

2. A new algorithm possessing computational complexity $O(1)$ to adaptively estimate the parameters of the Multifractal Lognormal Beta model;

3. A novel fuzzy flow rate control algorithm considering multifractal modeling and orthonormal basis functions that provides network performance improvement compared to others.

This paper is divided as follows: In Section 2, we describe the problem of network traffic flow rate control to improve QoS parameters, mainly regarding buffer occupation. We address the problem by considering fuzzy control techniques and multifractal modeling of traffic traces applied to 5G communications. Therefore, we first present in Section 3 a proposal of an algorithm to estimate the Lognormal Beta Model parameters in an adaptive manner. Next, in Section 4, concepts of orthonormal basis function, fuzzy logic and a proposal of flow rate control named GOBF-Fuzzy Flow Rate Control algorithm are presented. Regarding the wireless communication part of our work, in Section 5, we describe the 5G Downlink system (based on the first recommendations [1, 12]). In Section 6, we present the results obtained in the simulations with the considered traffic control algorithms. Finally, in Section 7, we conclude this work.

## 2 Problem description

The flow rates that arrive in a base station of mobile systems can be store in queues until data rates are provided. If wireless communication channel quality is low, low data rates are provided and can increase the buffer occupancy, causing data loss rate. In order to avoid data loss rate and to improve Qos parameters of the mobile network such as delay and buffer occupancy, flow rate control algorithms can be employed.

A flow rate control algorithm can be applied to the scenario of the Fig 1, representing a 5G downlink whose standard is described in [12] [1]. This control system aims to adaptively predict the queue size in the buffer and from the parameters of the traffic prediction model, to control the source rate in order to minimize the waiting time in the queue.

In the Fuzzy Flow Rate Control Scheme of Fig 1, the following variables are considered:

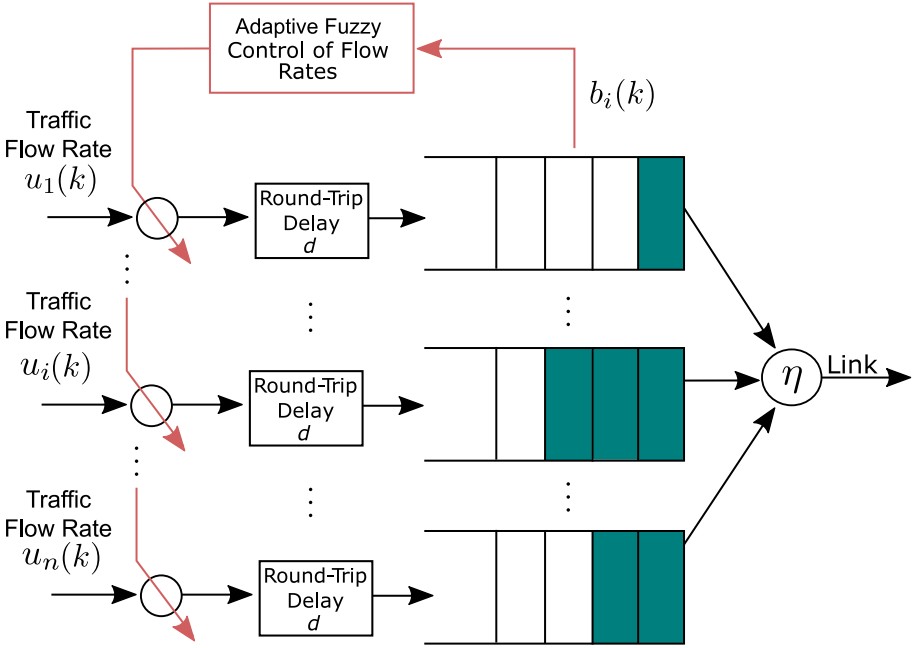

**Fig 1. Fuzzy flow rate control scheme.**

- Traffic flow rate $\mu_i(k)$ for the user $i$;

- Round-Trip delay $d$;

- Queue size $b_i(k)$ at the time instant $k$ for the user $i$;

- Link capacity $\eta$ of the system output;

- Desired Level for queue size $b_i^\tau$ for the user $i$.

The control scheme proposed in [24] accomplishes adjustment of flow rates aiming to individually maintain a desired queue length in the buffer to each source. This individual control makes the user enjoy the remaining bandwidth without knowledge of the queue size of the other sources. In this work, a new model for flow rate control considering Generalized Basis Functions for the fuzzy modeling is proposed that takes into account traffic from other sources of the system.

The proposed fuzzy traffic control scheme aims to take into account the round-trip delay by predicting the buffer occupancy behavior in order to avoid occurrence of congestion. To this end, we propose an adaptive algorithm to predict queue size in the buffer based on the past and present information of the source traffic rates. In addition, to obtain a waiting time (delay) in the queue as lower as possible, the proposed optimum control rate (see section 4.2) is applied to regulate the source rate $\mu_i(k)$. In this way, it is possible to confine the user delays within the required levels.

Due to different types of services and applications, such as data, voice and video being multiplexed in the nodes of the networks, the buffer occupancy dynamics is a complex and non-linear process, being an additional motive for the use of fuzzy systems. In this sense, we relate the optimum control rate to the adaptively computed parameters of the proposed GOBF Fuzzy model.

In order to present all the components of the proposed fuzzy control algorithm, first we propose in the next section an algorithm to adaptively estimate the parameters of the Lognormal Beta multifractal model in order to obtain the autocorrelation values to be used in the control algorithm.

## 3 Multifractal traffic analysis

Multifractal models can describe network traffic traces presenting long-range dependence, self-similarity and different scaling laws [25]. A stochastic process $X(k)$ is called multifractal if its increments $Z(k)$ satisfy:

$$\log E[|Z^{(m)}|^q] = \tau_0(q) \log m + \log c(q), \quad q > 0. \tag{1}$$

for $q \in Q$, where $Q$ is an interval on the real line, and $\tau(q)$ and $c(q)$ are functions with domain $Q$, $\tau(q)$ is the scaling function and $c(q)$ is the moment factor of the multifractal process.

One of the most important multifractal models present in the literature is the Lognormal Beta (LB) Model [10] that can be seen as a variation of the MWM (Multifractal Wavelet Model) [26]. The LB model consists of a multiplicative cascade process where a Beta distribution is considered for the cascade multipliers and the Lognormal distribution for the mass initialization [10].

Multiplicative cascades are recursive processes that can be used to generate multifractal processes [27]. The generation of a multiplicative cascade consists of the following steps: At stage $k = 1$, a mass with initial measure is divided, being multiplied by two random variables $r$ and $1 - r$, generating other two new masses, as depicted by Fig 2. The variable $r$ is named

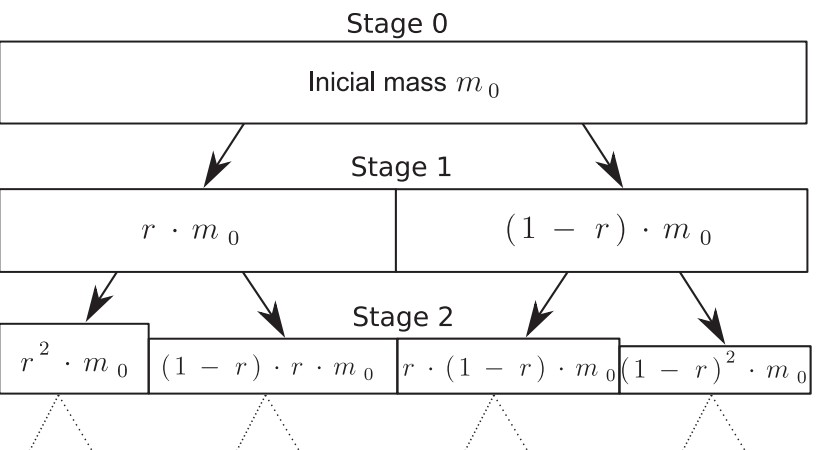

**Fig 2. Example of multiplicative cascades.**

multiplier. At the next stage, the masses are divided again by multipliers, repeating the recursive process until the desired number of stages are reached.

By allowing the cascade multipliers to be independent random variables in [0, 1] with probability density $f_R(X)$ and the initial mass measure generation with probability density $f_Y(X)$, a more general structure than the deterministic one, in which the multipliers and initial mass generation are fixed values, is obtained. In this way, the obtained process $\{\mu(\Delta t_k)\}_{k=1}^{2N}$ has in the stage $i$ of the cascade and in the dyadic interval of length $\Delta t_k = 2^{-k}$, starting in $t = 0.\eta_1 \ldots \eta_k = \sum_{i=1}^{k} \eta_i 2^{-i}$, the following measure:

$$\mu(\Delta t_k) = Y \cdot R(\eta_1) \cdot R(\eta_1, \eta_2) \ldots R(\eta_1, \ldots, \eta_k) \tag{2}$$

where $R(\eta_1, \ldots, \eta_i)$ is the cascade multiplier of the $i$-th stage. Since the multipliers $R(\eta_1, \ldots, \eta_i)$ are i.i.d. (independent and identically distributed), it can be shown that the measure $\mu$ satisfies the scale relation [10]:

$$E(\mu(\Delta t_k)^q) = E(Y^q) \cdot (E(R)^q)^k = E(Y^q) \cdot \Delta t_k^{-\log_2 E(R^q)} \tag{3}$$

According to [10], the multiplicative cascade process can generate a multifractal process since (1) can be approximated by (3). For this, the variables $Y$ and $R$ must meet the following equations:

$$\tau(q) = -\log_2 E(R^q) \tag{4}$$

$$c(q) = E(Y^q) \tag{5}$$

The scaling function $\tau(q)$ can be precisely modeled by assuming that $R$ is a random variable in [0, 1] with symmetric beta distribution $Beta(\alpha, \alpha)$ with $\alpha > 0$ [10]. Thus, we have:

$$\tau_0(q) = log_2 \frac{\Gamma(\alpha)\Gamma(2\alpha + q)}{\Gamma(2\alpha)\Gamma(\alpha + q)} \tag{6}$$

where $\Gamma(.)$ is the Gamma function and $\tau_0(q) = \tau(q) + 1$.

The random variable $Y$ is chosen in [10] as being the Lognormal with parameters $\rho$ and $\gamma$, where its $q$-th order moment is given by:

$$E[Y^q] = e^{\rho q + \gamma^2 q^2/2} \tag{7}$$

In summary, given the average $E[X(k)]$, the variance $var[X(k)]$ and the scaling function $\tau(q)$ of the network traffic flow, the parameters $(\rho, \gamma, \alpha)$ of the LB model can be determined by the following equations [10]:

$$c(q) = e^{\rho q + \gamma^2 q^2/2} 2^{N(q - \tau_0(q))} \tag{8}$$

$$E[X(k)] = e^{\rho + \gamma^2/2} \tag{9}$$

$$var[X(k)] = e^{2\rho + 2\gamma^2}\left(\frac{\alpha + 1}{\alpha + 1/2}\right)^N - e^{2\rho + \gamma^2} \tag{10}$$

where $N$ is the stage number in the multiplicative cascade process. Notice that in order to obtain the parameters $\rho$, $\gamma$ and $\alpha$, the calculation of $\tau_0(q)$ and $c(q)$ is necessary. Adaptive approaches for estimating $c(q)$ and $\tau(q)$ are addressed in **Propositions 1** and **2**.

**Proposition 1**. *The moment factor $c(q)$ of a traffic flow can be adaptively computed in function of the moments of the aggregated increment process $Z^{(m)}$ in the scale $m = 1$ by*:

$$c(q) = \frac{kE[|Z^{(1)}|^q](k)}{(k+1)} + \frac{Z(k+1)^q}{k+1} \tag{11}$$

*Proof.* Once $E[|Z^{(1)}|^q](k) = (1/k)\sum_{i=1}^{k} Z_i^q$, the adaptive average operation for the subsequent time instant $(k+1)$ is obtained by:

$$E[|Z^{(1)}|^q](k+1) = \frac{1}{k+1}\frac{k}{k}\sum_{i=1}^{k} Z_i^q + \frac{1}{k+1}Z(k+1)^q \tag{12}$$

Replacing $E[|Z^{(1)}|^q](k)$ in (12) and substituting $m = 1$ in (1), we obtain the moment factor $c$ ($q$) as we wanted to demonstrate.

**Proposition 2**. *The scaling function $\tau(q)$ of a network traffic flow $X(k)$ can be given in function of the moments of the increment process aggregated on scale $m$ by*:

$$\tau_0(q) = \frac{\log E[|Z^{(m)}|^q] - E[|Z^{(1)}|^q]}{\log m} \tag{13}$$

*where*:

$$E[|Z^{(m)}|^q](k+1) = \frac{\lfloor k/m \rfloor}{\lfloor (k+1)/m \rfloor}\mathcal{M} + \frac{\mathcal{E}^q}{\lfloor (k+1)/m \rfloor} \tag{14}$$

$$\mathcal{M} = \begin{cases} \mathcal{M}, & if \ \ \mathrm{mod}\ (k, m) \neq 0 \\ E[|Z^{(m)}|^q](k), & if \ \ \mathrm{mod}\ (k, m) = 0 \end{cases} \tag{15}$$

$$\mathcal{E} = \begin{cases} \mathcal{E} + Z(k), & if \ \ \mathrm{mod}\ (k, m) \neq 0 \\ Z(k), & if \ \ \mathrm{mod}\ (k, m) = 0 \end{cases} \tag{16}$$

*Proof.* The average of the aggregated process $Z^{(m)}$ at time instant $k$ considering a complete time window of aggregation, can be given by:

$$E[|Z^{(m)}|^q](k) = \frac{1}{\lfloor k/m \rfloor} \sum_{i=1}^{\lfloor k/m \rfloor} Z_i^{(m)} \tag{17}$$

In order to calculate the average of the real aggregated process, it is necessary to consider data in complete windows of size $m$. Therefore, for time interval different from a complete window, the average can be written as function of a complete window plus an estimate $\mathcal{E}$ of the data before completing a new window, that is:

$$E[|Z^{(m)}|^q](k+1) = \frac{\lfloor k/m \rfloor E[|Z^{(m)}|^q](k)}{\lfloor (k+1)/m \rfloor} + \frac{\mathcal{E}^q}{\lfloor (k+1)/m \rfloor} \tag{18}$$

In order to update the average $E[|Z^{(m)}|^q](k)$, the variable $\mathcal{M}$ is created. That is, at time instant $mod(k, m) = 0$, $\mathcal{M}$ is updated with the average of a complete window according to (15). The estimation $\mathcal{E}$ is updated at each time instant with the current increment process $Z(k)$, when a complete time window is reached ($mod(k, m) = 0$), $\mathcal{E}$ is reinitialized with $Z(k)$ according to (16). Replacing $\mathcal{M}$ in (18), we obtain (14), as we wanted to demonstrate.

Substituting (9) into (1), the parameters $\rho(k)$ and $\gamma(k)$ can be written as:

$$\rho(k) = 2\log(E[X(k)](k)) - \frac{\log c(2)}{2} + \frac{N}{2}[2 - \tau(2)]\log 2 \tag{19}$$

$$\gamma(k) = \sqrt{2\log E[X(k)](k) - 2\rho(k)} \tag{20}$$

From (10), we can directly estimate the parameter $\alpha$ by:

$$\alpha(k) = \left(\sqrt[N]{G} - 2\right) \cdot \left(2 - 2\sqrt[N]{G}\right)^{-1} \tag{21}$$

where $G = e^{(-2\rho(k) - 2\sigma(k)^2)} \cdot (var[X(k)](k) + E[X(k)](k)^2)$.

**Propositions 1** and **2** in conjunction with (19), (20) and (21) allow us to adaptively estimate the LB parameters ($\rho, \gamma, \alpha$) with an optimum computational complexity of $O(1)$. Once estimated the parameters ($\rho, \gamma, \alpha$), the autocorrelation function values of the process $X(k)$ can be computed by [10]:

$$r(x) = e^{2\rho + \gamma^2} \cdot \frac{\alpha(\alpha + 1)^{N-1}}{(\alpha + 1/2)^N} \cdot x^{-\log_2\left(\frac{\alpha+1}{\alpha+1/2}\right)} \tag{22}$$

Notice that we propose to compute the autocorrelation values via a multifractal model instead of direct from the traffic process in order to provide a complete model based control. Besides, one can predict some network performance factors only by analysing the variation of the modeling parameters.

## 4 Generalized OBF fuzzy modeling and control

Orthonormal basis functions have arisen with the principle of searching for alternatives to express the transfer function of a system, becoming possible to reduce the number of system inputs and to increase process modeling performance [3]. Among the known bases, we can highlight the Laguerre basis and the Generalized basis [3]. The generalized basis function is

given by:

$$f_i(q) = \frac{\sqrt{1 - |p_i|^2}}{q - p_i} \prod_{j=1}^{i-1} \frac{(1 - p_j^* q)}{(q - p_j)} \tag{23}$$

where $\{p_j: j = 1, 2, 3, \ldots\}$ is an arbitrary sequence of poles that satisfies: $p_j \in \mathbb{C} : |\beta_i| < 1$. The poles can be obtained by the Levinson-Durbin recursion [28].

The output of the orthonormal basis function models can be written as $y(k) = \mathcal{H}(l_1(k), \ldots, l_n(k))$, where $l_i(k) = f_i(q)u(k)$ is the $i$-th basis function at time instant $k$ and $\mathcal{H}$ is a non-linear operator. In this work, due to its modeling capability, we consider to apply a TSK fuzzy system to model the OBF operator $\mathcal{H}(.)$ [29]. In the next section, we describe the LMS Fuzzy algorithm since it is used to update the parameters of the proposed GOBF Fuzzy Traffic Model as well as the optimal control rates (section 4.2).

## 4.1 LMS fuzzy algorithm

Consider the input vector $[x(k)]$ with $x(k) \in \mathbb{U} \equiv [C_1^-, C_1^+] \times [C_2^-, C_2^+] \ldots [C_n^-, C_n^+] \subset \mathbb{R}^n$ where $\mathbb{U}$ is the set of filter input samples, $\mathbb{R}$ is the output set and $[d(k)]$ the desired response, where $k = 0, 1, 2, \ldots$ is the time instant and $C_i$ is the boundary of the interval $[C_1^-, C_1^+]$.

In order to obtain the fuzzy system output $f_k(x) : \mathbb{U} \subset \mathbb{R}^n \to \mathbb{R}$, we must minimize the following square error expression:

$$L = E[(d(k) - f_k(x(k)))^2] \tag{24}$$

where $f_k(x)$ minimizes (24).

The LMS Fuzzy algorithm design is described by the following steps:

- **Step 1:** Define $M$ fuzzy sets $F_i^l$ for each interval $[C_i^-, C_i^+]$ of the input space $\mathbb{U}$, with the membership function given by:

$$\mu_{F_i^l} = exp\left[-\frac{1}{2}\left(\frac{x_i - \bar{x}_i^l}{\sigma_i^l}\right)^2\right] \tag{25}$$

where $x = (x_1, \ldots, x_n)^T \in \mathbb{U}$, $\bar{x}_i^l$ and $\sigma_i^l$ are the mean and the standard deviation of the Gaussian membership function $\mu_{F_i^l}$

- **Step 2:** Build the fuzzy rule set IF-THEN by the following statement:

$$R^l = \begin{cases} \text{Se } x_1 \text{ is } F_1^l \text{ and } \ldots \text{ and } x_n \text{ is } F_n^l \\ \text{then } d \text{ is } G^l \end{cases} \tag{26}$$

where $d \in \mathbb{R}$ and $F^l$ is defined in step 1, with membership function $\mu_{F_i^l}$.

- **Step 3:** Obtain the fuzzy system output value, that is equivalent to equivalent to the fuzzy basis function (FBF) [29], by the following equation:

$$f_k(x) = \frac{\sum_{l=1}^{M} \theta^l (\prod_{i=1}^{n} \mu_{F_i^l}(x_i))}{\sum_{l=1}^{M} (\prod_{i=1}^{n} \mu_{F_i^l}(x_i))} \tag{27}$$

where $\theta^l$ is a weight parameter.

- **Step 4:** Using the LMS algorithm, update the parameters $\theta^l$, $\bar{x}_i^l$ e $\sigma_i^l$. The parameters $\theta^l$, $\bar{x}_i^l$ and $\sigma_i^l$ can be initialized by human knowledge (specialized) or be randomly selected. These parameters are adaptively updated at each iteration by the following equations [29]:

$$\theta^l(k) = \theta^l(k-1) + \delta[d(k) - f_k]\frac{z^l(k-1)}{g(k-1)} \tag{28}$$

$$\bar{x}_i^l(k) = \bar{x}_i^l(k-1) + \delta[d(k) - f_k]\frac{\theta^l(k-1) - f_k}{g(k-1)}z^l(k-1)\frac{x_i(k) - \bar{x}_i^l(k-1)}{\left(\sigma_i^l(k-1)\right)^2} \tag{29}$$

$$\sigma_i^l(k) = \sigma_i^l(k-1) + \delta[d(k) - fk]\frac{\theta^l(k-1) - f_k}{g(k-1)}z^l(k-1)\frac{(x_i(k) - \bar{x}_i^l(k-1))^2}{\left(\sigma_i^l(k-1)\right)^3} \tag{30}$$

where

$$z^l(k-1) = \prod_{i=1}^{n}exp\left[-\frac{1}{2}\left(\frac{x_i(k) - \bar{x}_i^l(k-1)}{\sigma_i^l(k-1)}\right)^2\right] \tag{31}$$

and

$$g(k-1) = \sum_{l=1}^{M}z^l(k-1) \tag{32}$$

for $l = 1, 2, \ldots, M$, $i = 1, 2, \ldots, n$ and $\delta$ is the learning rate that satisfies $0 < \delta < 1$.

## 4.2 Optimal control rate

In this section, as part of the proposed flow rate adaptive control scheme, we present an expression for the calculation of the optimal control rate $\mu(k)$ by minimizing the following cost function:

$$J(k+d) = E\left[\frac{b_i(k+d)}{\eta} + \frac{\lambda}{2}\left(\mu_i(k) - R_{i;min}\right)^2\right], \tag{33}$$

where $\lambda$ is a weighting factor and $\eta$ is the link capacity. The cost equation $J$ takes into account the waiting time in the buffer given by $b_i(k+ d)/\eta$. In order to comply minimum flow rates $R_{i;min}$ for user $i$, the second term is added to the cost function $J$.

**Proposition 3**. *Considering a downlink system with n users and $R_{i;min}$ the minimum rate for user i, the optimal control rate in terms of minimizing $J(k + d)$ in* (33) *is given by*:

$$\mu_i^o = R_{i:min} - \frac{(2 - f_k(x))}{\lambda\eta g(k-1)}\sum_{l=1}^{M}\frac{\theta^l(\mu_i - \bar{\mu}_i^l)z^l(k-1)}{\left(\sigma_i^l\right)^2} \tag{34}$$

*where $f_k(x)$ is the fuzzy system output, $\theta^l$, $\bar{x}_i^l$ and $\sigma_i^l$ are fuzzy model parameters, $z^l$ is given by* (31) *and $g(k-1)$ by* (32).

*Proof.* Deriving the function $f_k(x(k))$ (Eq (27)) in relation to $x_i$, we obtain:

$$\frac{\partial f_k(x)}{\partial x_i} = \frac{(2 - f_k(x))}{g(k-1)}\sum_{l=1}^{M}\frac{\theta^l(x_i - \bar{x}_i^l)z^l(k-1)}{\left(\sigma_i^l\right)^2} \tag{35}$$

The proposed optimal control rate is given as a function of the value of the queue size in the buffer $d$ steps ahead. An estimate of the queue size $b_i(k)$ in the buffer $d$ steps ahead is provided by the output of the proposed fuzzy predictor when applied to the prediction of samples of this process by (27), i.e. $f_k(x) = b_i(k + d)$. Deriving (33) in relation to $\mu$, we have:

$$\frac{\partial J}{\partial \mu} = \frac{\partial b_i(k + d)}{\eta \partial \mu} + \lambda(\mu - R_{i;min}) = 0 \tag{36}$$

Replacing (35) and $f_k(x) = b_i(k + d)$ in (36) and isolating $\mu$ we obtain the optimal control rate given by (34).

The proposed fuzzy traffic control algorithm that makes use of generalized orthonormal basis functions and the optimal control rate is presented in **Algorithm 1**.

**Algorithm 1**: **GOBF-Fuzzy Flow Rate Control**

**1:** Calculate $E[|Z^{(1)}|^q](k + 1)$ and $E[|Z^{(m)}|^q](k + 1)$ by (11) and (14).
**2:** Calculate $c(q)$ and $\tau(q)$ by (11) and (13).
**3:** Calculate $(\rho(k), \gamma(k), \alpha(k))$ by (19), (20) and (21).
**4:** Calculate the GOBF-poles by the Levinson-Durbin Recursion using the autocorrelation values given by (22).
**5:** Compute the GOBF Basis by (23) with the obtained poles.
**6:** Calculate the Optimal Control Rate by (34).
**7:** Update the GOBF Fuzzy model parameters $\theta^l$, $\bar{x}_i^l$ and $\sigma_i^l$ with the LMS algorithm.
**8:** Return to Step 1 until the end of iterations.

## 5 5G downlink system

The fifth generation of mobile communications is intended to provide higher transmission rates, lower latencies than the earlier generation and support for communications of a massive number of devices (Internet of Things—IoT). To this end, in 2017, the International Telecommunication Union (ITU) group established the following specifications for 5G technology: bandwidth up to 1GHz in the high frequency region (millimeter waves); Downlink transmission rate of at least 20Gbps; 10Gbps Uplink transmission rate; reduction in latency for values less than 1$ms$ and a device density of 1 million per square kilometer [30].

IMT-2020 requires high transmission rate for both Downlink and Uplink to 5G networks [31]. This high transmission rate can be achieved by two alternatives: to increase the spectral efficiency or to increase bandwidth. In order to increase the spectral efficiency, one can change the modulation and the coding. The earliest recommendations for 5G NSA (LTE Release 15 Non-Standalone Architecture) downlink transmission considers advances in the OFDM LTE for providing robust communication in frequency selective channels [12]. These first recommendations are based on CP-OFDM (Cyclic Prefix OFDM) used in LTE radios, on the carrier aggregation to increase the amount of available resources and on Multiple Input Multiple Output (MIMO) techniques. In this work, we consider the F-OFDM (Filtered Orthogonal Frequency-Division Multiplexing) as modulation technique. The F-OFDM is a variant of the OFDM technique that makes flexible the allocation of bandwidth by varying the number of F-OFDM subcarriers used for transmission and reduces out-of-band (OOB) emissions compared to traditional OFDM [32].

Given the high data rates required to 5G technology, the coding for data transmission is based on the Low-Density Parity-Check (LDPC) [12]. LDPC codes are attractive from the viewpoint of implementation; especially at higher code rates, due to their lower complexities than those of the Turbo codes, which is used in 4G [33].

The second alternative to provide higher transmission rate is to increase the bandwidth. Due to frequency spectrum occupancy by other technologies, it is difficult to obtain

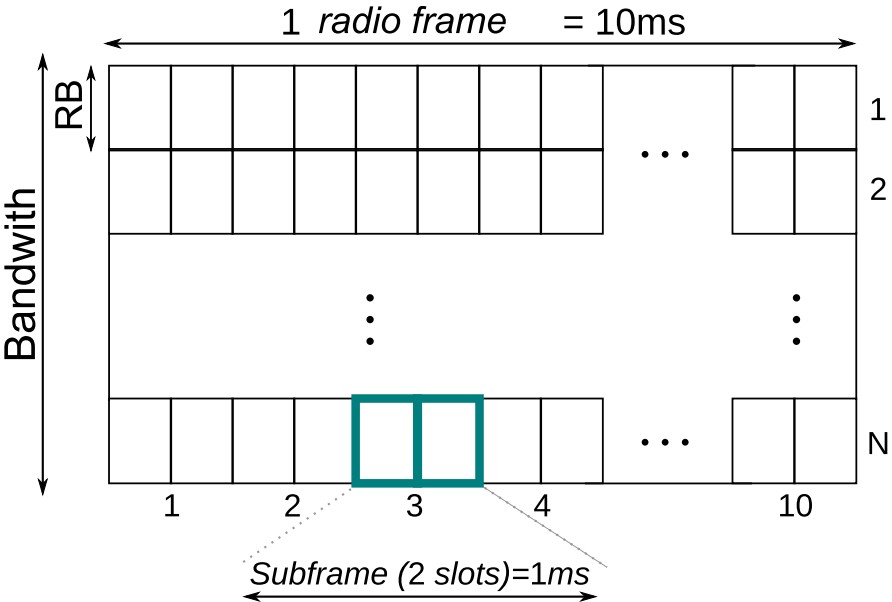

**Fig 3. 5G basic time-frequency resource structure ($\mu = 0$).**

bandwidths around the order of 1GHz, as required by 5G technology. This is the reason for using millimeter waves whose frequencies are between 6GHz and 100GHz [33, 34].

Another requirement for 5G networks is to decrease latency to values less than 1ms. For LTE (releases from 8 to 14), the shortest allocation time for users is 1ms. In order to decrease latency, the first specifications of 5G networks establish the following decision times for scheduling resources: 1ms, 0.5ms, 0.25ms, 0.125ms and 0.0625ms. With shorter times, it is possible to reduce the delay.

In the downlink of LTE-based networks, each terminal reports its instant estimate of channel quality to the base station. This estimate is obtained by measuring a reference signal sent by the base station. Based on the channel quality estimate, the scheduler can assign resources to users. A combination of resource blocks can be assigned to each terminal at the scheduling time interval. Resource blocks consist of 12 subcarriers in one slot. Scheduling decisions are made at each scheduling interval in the time domain. The Scheduling Block (SB) is defined as a pair of resource blocks [1].

The basic time slot in LTE systems has six or seven OFDM symbols, depending on the use of normal or extended cyclic prefix. Fig 3 shows the 5G basic time-frequency resource structure. For the LTE Release 15 NSA, in order to decrease the latency time of the system, there are 5 possible scheduling times [31]. The scheduling times are defined according to the Table 1, where each time is classified by a number $\mu$, named numerology. Besides scheduling time,

**Table 1. Numerology for the LTE Release 15 NSA [1].**

| $\mu$ | Time ($1ms/2^{\mu}$) | $\Delta f = 2^{\mu} \cdot 15[KHz]$ | Ciclic Prefix |
|---|---|---|---|
| 0 | 1ms | 15 | Normal |
| 1 | 0.5ms | 30 | Normal |
| 2 | 0.25ms | 60 | Normal, Extended |
| 3 | 0.125ms | 120 | Normal |
| 4 | 0.0625ms | 240 | Normal |

**Table 2. Resource Block (RB) bandwidth for the LTE Release 15 NSA [1].**

| $\mu$ | RB Bandwidth |
|---|---|
| 0 | 180Khz |
| 1 | 360Khz |
| 2 | 720Khz |
| 3 | 1440Khz = 1.44Mhz |
| 4 | 2880Khz = 2.88Mhz |

numerology defines the entire OFDM structure according to the spacing between the subcarriers and number of symbols per block of resources.

In Fig 3, each Resource Block (RB) corresponds to a time slot. In the frequency domain, this resource block has a bandwidth equal to $180 kHz$ for $\mu = 0$, where 12 subcarriers of $15 kHz$ are grouped together. Two resource blocks form the Scheduling Block (SB), thus having the duration time of $1 ms$ [31] [12]. For other numerologies the bandwidths can be seen in Table 2.

In the 5G downlink transmission system, scheduling blocks are allocated to perform data transmission between the 5G eNodeB and the user equipment. The transmission rate is directly proportional to the channel quality. The structure of the 5G downlink allows the choice between some modulations and coding scheme (MCS) with the purpose of optimizing the transmission according to the channel quality. In the 5G network, the CQI (Channel Quality Indicator) index dictates the code rate and modulation scheme that will be used.

According to the 3GPP (3rd Generation Partnership Project) specification, in each Transmission Time Interval (TTI), at most one transport (scheduling) block of a certain size is transmitted over the radio interface [35]. There is a Transport Format (TF) associated with each transport block, specifying how the transport block is to be transmitted over the radio interface. The transport format includes information about the transport-block size, the modulation and code scheme (MCS), and the antenna mapping. The set of modulation schemes supported in the 5G downlink includes QPSK, 16-QAM, 64-QAM and 256QAM, corresponding to two, four, six and eight bits per modulation symbol, respectively [1]. By varying the transport format, it can be achieved different data rates.

## 6 Results and discussions

In this section we present the results and discussions about the simulations of the proposed fuzzy flow rate control, but first we analyze the adaptive multifractal parameters estimation performance.

Simulations related to the estimation of multifractal parameters with different network traffic traces were carried out. However, in order to be concise, we present in this paper the results for one of them that represents the major observed behaviour. The chosen network traffic trace was the MAWI-201804011400 [36], in this paper named MAWI. The MAWI traces represent daily data traffic of different applications from the collected at the Internet backbone of the Measurement and Analysis on the WIDE Internet (MAWI) working group. In this paper, the MAWI traces were chosen since they correspond to recent and modern wireless network traffic flows [37, 38].

In order to verify the accuracy of the proposed recursive equations that compose the algorithm to adaptively estimate the LB Multifractal model parameters ($\rho, \gamma, \alpha$), we compare their values to those provided by the on-batch estimation method. Fig 4 presents the values for the $\rho$ adaptively calculated using (19) and by the on-batch method using all samples of the MAWI network traffic. The on-batch method consists of computing the moment factor $c(q)$ and

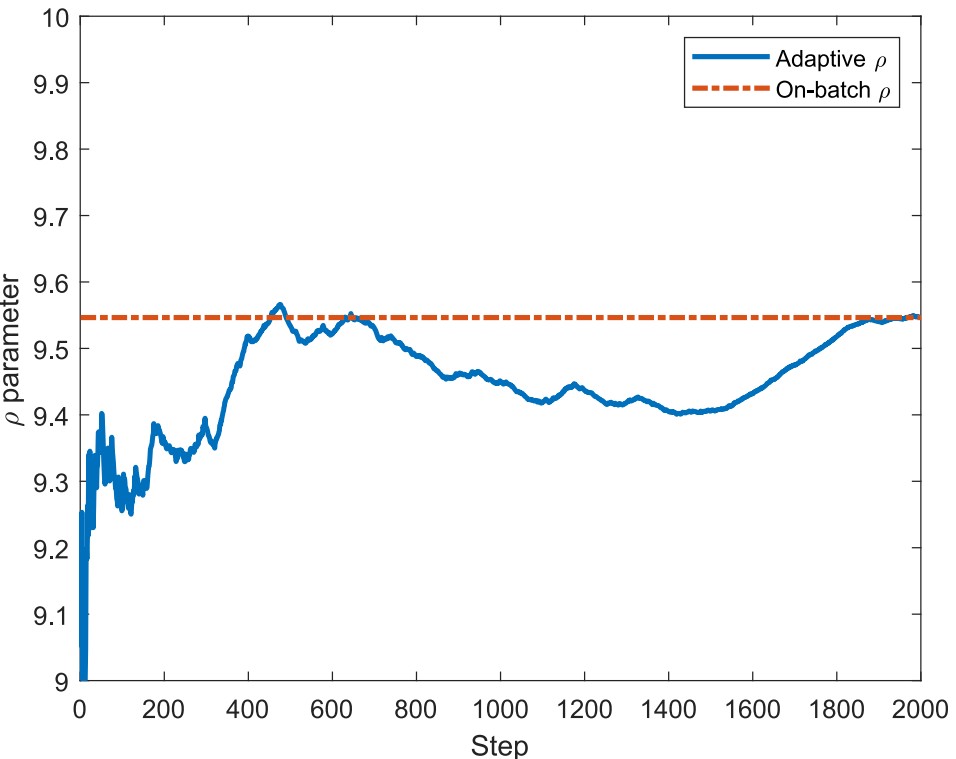

**Fig 4. Adaptive and on-batch estimation of $\rho$ for 2000 samples from the MAWI Trace aggregated in 1*ms*.**

scaling function $\tau(q)$ of the entire process $X(k)$ all at once and obtaining the parameters $\rho$, $\gamma$ and $\alpha$. It can be noted that the final values of $\rho$ adaptively estimated are similar to those of the non-adaptive method at the end of the iterations, with a percent error of $5.63 \cdot 10^{-4}$%. Figs 5 and 6 show that the estimation of $\gamma$ and $\alpha$ achieves similar convergence behaviour to that presented by the estimation of $\rho$ in Fig 4, presenting a percent error of $0.97 \cdot 10^{-2}$% and $1.82 \cdot 10^{-12}$%, respectively.

Regarding the simulation of a scenario towards 5G, its characteristics are described as follows: The Clustered Delay Line (CDL) channel model was chosen to simulate a 5G configuration with MIMO 8x8 antennas and carrier frequency of 26 GHz according to [1]. We consider two scenarios: The first one is for a channel with a bandwidth of 400MHz and the second one is for carrier aggregation of 3 carriers with a bandwidth of 400MHz (1.2GHz). Both 5G scenario structures is formed by considering numerology $\mu = 3$, with subcarrier spacing of 120KHz, 6 OFDM symbols per slot, 264 Resource Blocks per carrier, 5G eNodeB Transmission Power of 44dBm, eNodeB Gain Antenna of 15dBi, User Antenna Gain of 0dBi and Low Density Parity Check (LDPC) for the channel coding [1]. The choice of numerology $\mu = 3$ was due to the fact that it is the value that represents the highest transmission rate among other numerologies. Besides, numerology $\mu = 4$ is used only for signaling and control [31]. As modulation technique, we considered the F-OFDM with a length of 512 subcarriers in the simulations. The simulations represent the downlink transmission in a unique cell of a multicarrier 5G system.

We compare the performance of the proposed fuzzy control to those of the Proportional Flow Rate Control [16] and GCC (Google Congestion Control) [19] and a similar control but

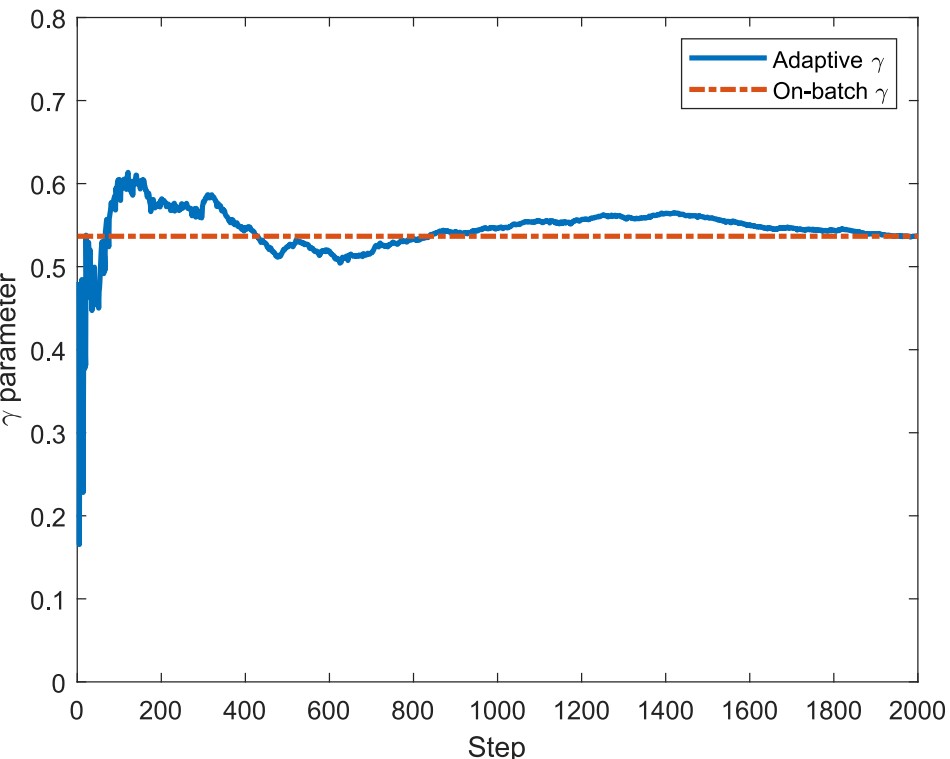

**Fig 5. Adaptive and on-batch estimation of $\gamma$ for 2000 samples from the MAWI Trace aggregated in 1$ms$.**

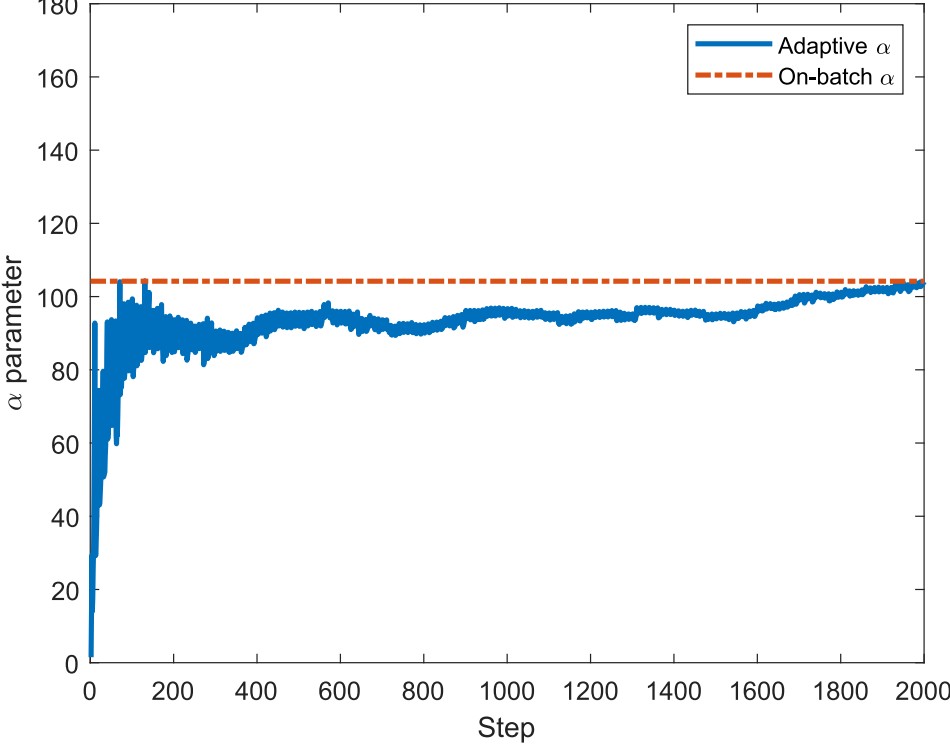

**Fig 6. Adaptive and on-batch estimation of $\alpha$ for 2000 samples from the MAWI Trace aggregated in 1$ms$.**

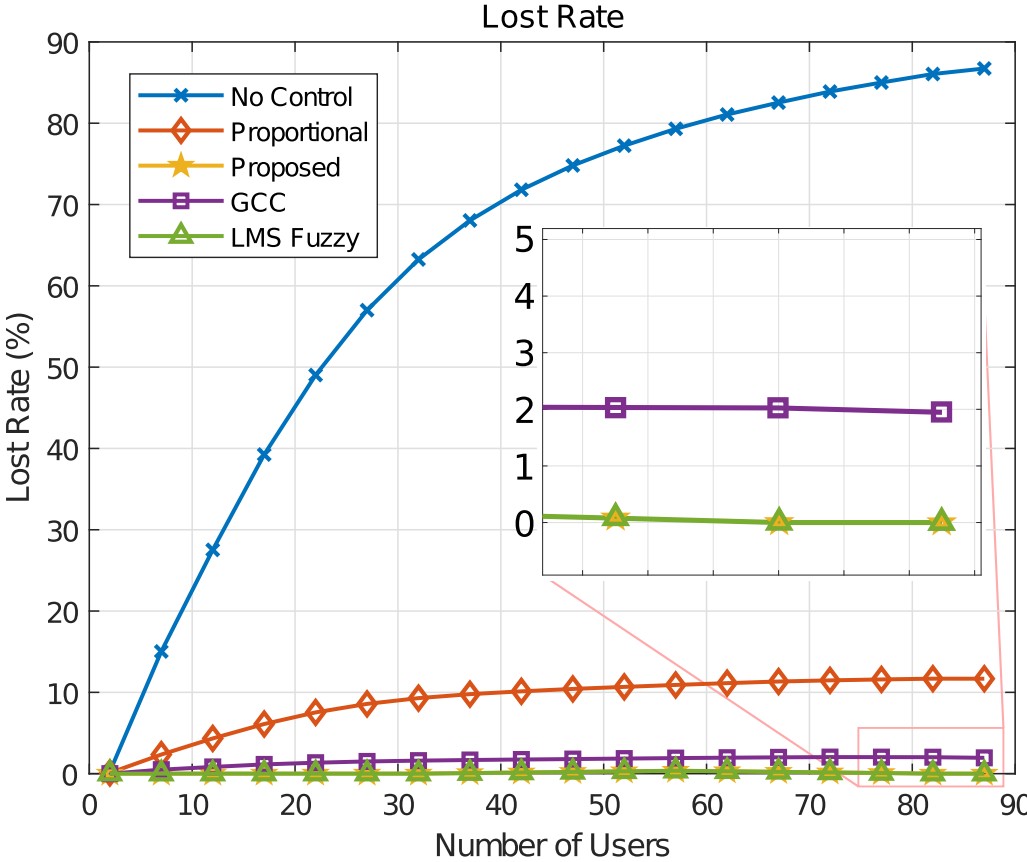

**Fig 7. Byte loss rate (%).**

using a LMS Fuzzy training algorithm. The Proportional Flow Rate Control monitors the queue sizes and uses this value to regulate the flow rate. Let $b^\tau$ be the control reference value of the queue size in the buffer in eNodeB. In the Proportional Flow Rate Control, if the queue size is below a minimum threshold, there is an increase of 5% in the flow rate. If the queue size is between a minimum and a maximum threshold, the flow rate is proportionally increased. Finally, if the queue size is above a maximum threshold, there is a decrease in flow intensity of 5%. In this work, we considered a buffer size of 60 kB (representing a practical value) for each user and the desirable size $b^\tau$ equal to 40% of the total size, that is, $b^\tau = 20$ kB.

The GCC algorithm control network traffic flows according to the minimum value between two rates [19]. The first rate is regulated according to the amount of data lost in the transmission. That is, if the loss rate is lower than a desired value, an increase in the flow rate occurs, if the loss rate is higher than the desired value (2%), the flow rate is decreased. If the loss rate is between the desired minimum and the maximum, no change in flow rate occurs. The second rate refers to the buffer occupancy. Fig 7 shows the byte loss rate values for the control algorithms considered in the simulations. The lowest loss rate values (zero values) were obtained by the proposed fuzzy approaches: LMS-Fuzzy Control and the GOBF-Fuzzy Control algorithms. We emphasize that what we call LMS-Fuzzy Control consists of the proposed control approach but without considering the multifractal based OBF modeling. Notice that the GCC algorithm presents values of loss rate close to the GCC desired value of 2%. It can be noted that, although providing a reduction in the loss rate in relation to the method without flow

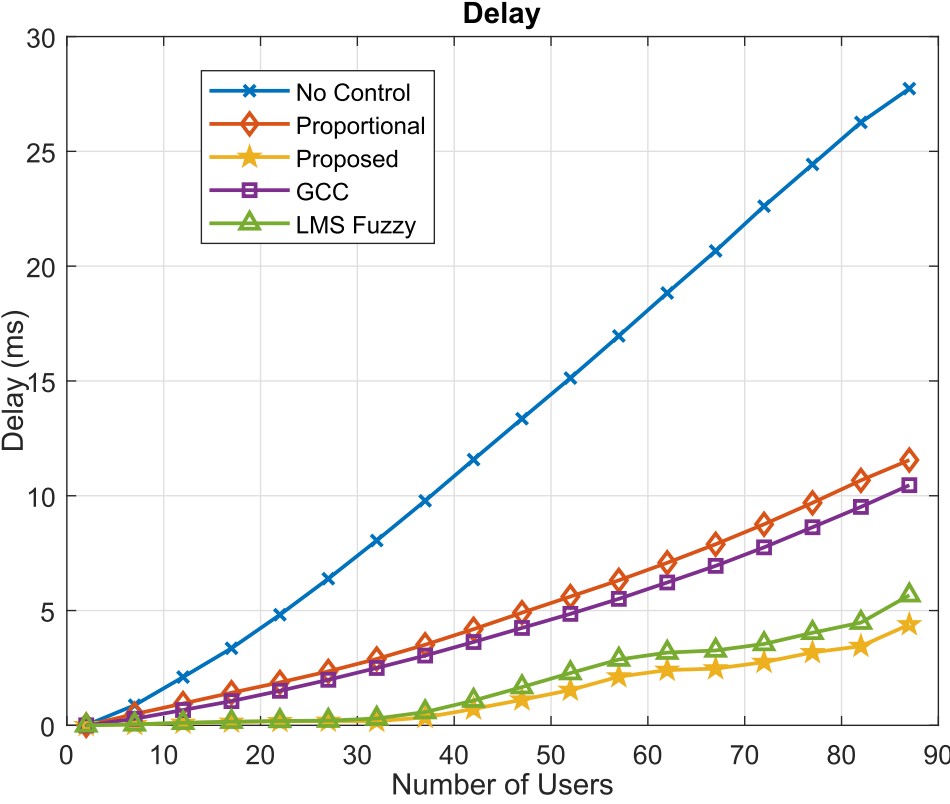

**Fig 8. Delay (*ms*).**

control, the Proportional algorithm presents loss rates close to 10%, which in practice can be considered a relatively high value.

Another important QoS parameter to be analyzed is network delay. Fig 8 presents the waiting times (Delay) of data in the queue for the considered algorithms. In this case, we can observe that the proposed GOBF-Fuzzy control provided lower delay values than the proposed LMS-Fuzzy control and the others, but similar to that of the GCC algorithm. The delay is directly related to the buffer occupancy, so the higher the buffer occupancy, the longer the delay. In Fig 9, the results of the buffer occupancy (%) are presented. It is noted, that the values of buffer occupancy presented by the proposed algorithms were the smallest, according to the behaviour present by the delay values in Fig 8.

The flow control performance is an important factor for the communication quality of mobile networks. Incoming data can be stored in queues and if the buffer occupancy is increasing too much, data loss can occur and communication may be inefficient, requiring data retransmission and reducing the effective transmission data rate of users, which leads to lower baud rates than those stipulated in the contract with the operator. The results of the simulations carried out indicate that it is possible to obtain lower buffer occupancy values by using the proposed network traffic flow control algorithm compared to the others.

We also evaluate the throughput per user provided by the control algorithms. According to Fig 10, we can state that the throughput values per user provided by the proposed fuzzy control algorithms are higher than those of the Proportional. That is, it can be observed that the proposed fuzzy approaches are capable of maintaining low delay and loss rate values to the system

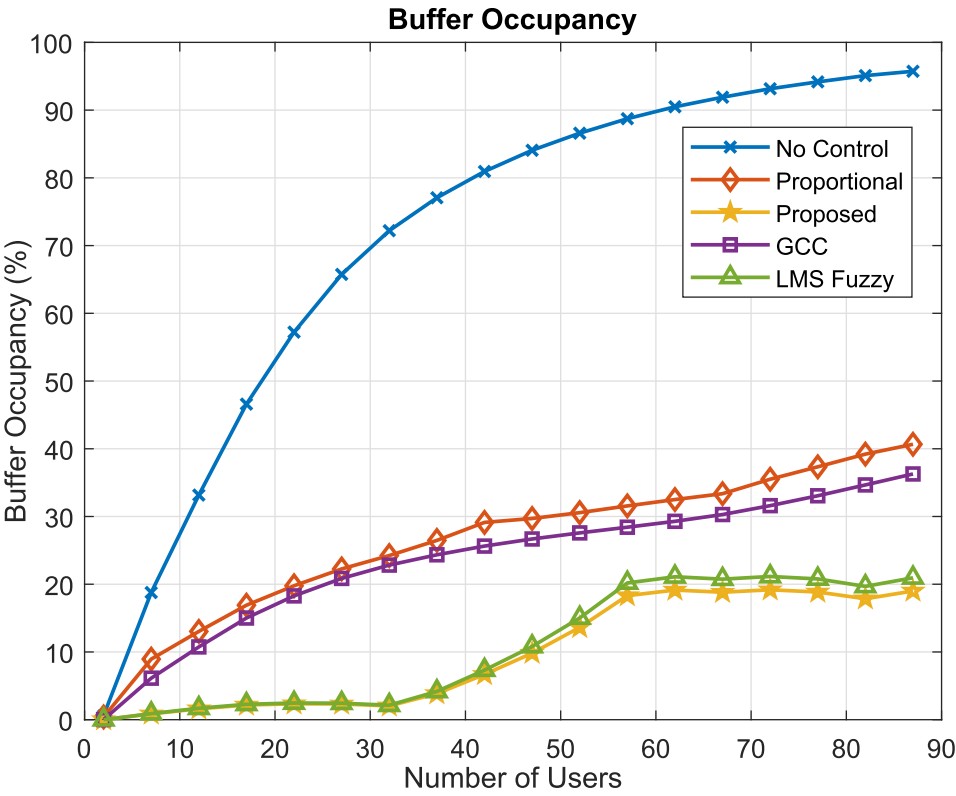

**Fig 9. Buffer occupancy (%).**

downlink without penalizing so much the throughput per user compared to the proportional case. Moreover, when compared to the GCC, the proposed algorithm presented slightly inferior performance in terms of throughput values, although it has resulted in a better performance for delay and buffer occupancy parameters. Notice also that the minimum flow rates $R_{(\forall i;min)}$ chosen to be 100Mbps in the parameter setting of the fuzzy algorithms was attained to all users in the simulations.

The delay in the network is an important QoS parameter that influences the number of users that can be allowed in the wireless system under the contract with the service provider. Let us assume, for example, a limiting value of 5*ms*. That is, data communication in the considered wireless network scenario should not exceed this value. It can be seen from Fig 8 that the proposed flow control algorithm can accommodate up to 87 users in the system, while the GCC control algorithm can allocate 52 users in the system; a difference of 35 users, that represents a significant financial loss for the mobile operator. By considering the case without flow control, it can be accommodated 22 users and 47 users using the proportional control algorithm and 82 users using the LMS Fuzzy algorithm, representing financial losses of 65, 40 users and 5, respectively.

Fig 11 presents the results of the 5G simulation link utilization for the flow control algorithms. It should be noted that the control algorithms did not use all the available rate, thus being able to control the user's rate to improve the quality of the system in relation to the other parameters of quality of service. It should also be noted that the proportional algorithm presented the lowest link utilization values, while the other algorithms were more opportunistic in the link utilization of the downlink 5G.

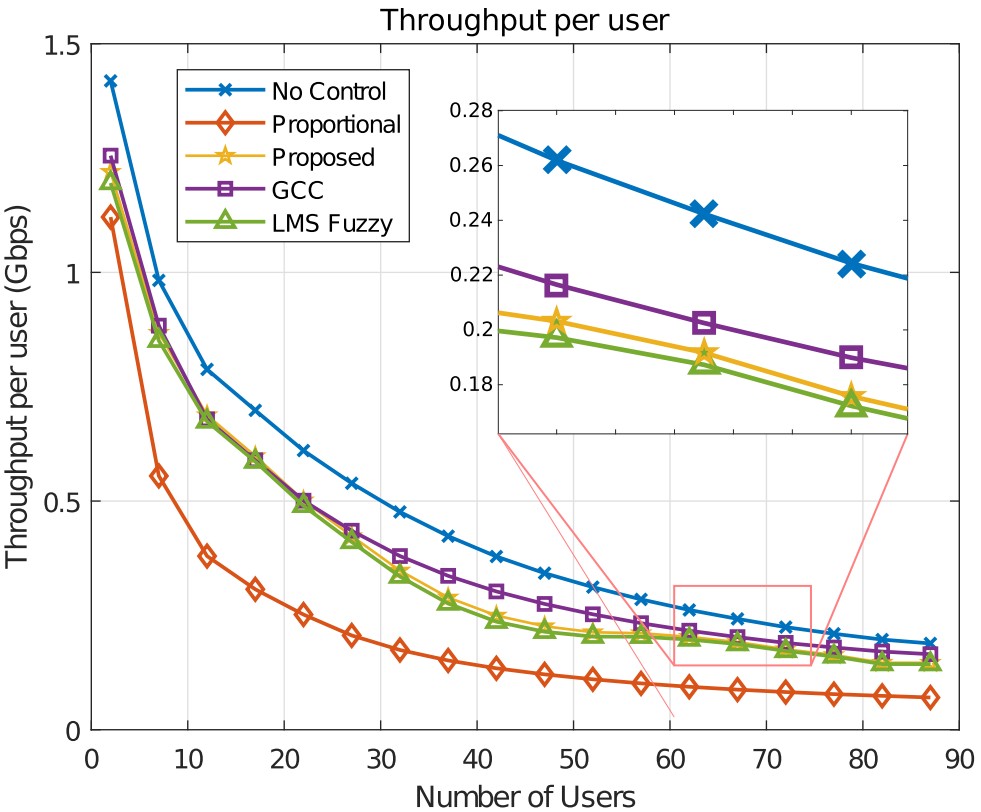

**Fig 10. Throughput per user (*Mbps*).**

In resume, the simulation results show that the GOBF-Fuzzy Control can provide gains to the network performance in terms of buffer occupancy, loss rate and delay compared to other control algorithms and a throughput per user almost equal to that without control. Simulations considering various other traffic traces and scenario configurations were carried out, providing similar performance results to those presented in this paper. In order to attain better network performance, the OBF-Fuzzy requires a slightly more computational complexity than the LMS Fuzzy Control, being $O(nM + P^2)$ against $O(nM)$ of the LMS Fuzzy, where $n$ is the number of inputs, $M$ is the number of fuzzy rules and $P$ the number of poles.

## 7 Conclusion

In this work, we present a fuzzy flow control system applied to a initial 5G downlink system (LTE Release 15 NSA) considering the F-OFDM modulation technique. To this end, we propose an equation to calculate the optimal traffic flow rates for users in terms of minimizing delay and guaranteeing a minimum rate to them. We concluded that by inserting generalized orthonormal basis functions derived from multifractal modeling into the fuzzy system, we can enhance network control performance. In fact, it could be observed that the GOBF fuzzy control provided better results in terms of loss rate, buffer occupancy and delay than the other considered algorithms.

The proposed algorithm provided better simulation results in terms of delay, buffer occupancy and loss rate than the other considered algorithms. The enhanced performance of the proposed fuzzy control algorithm is obtained at the cost of increasing the computational

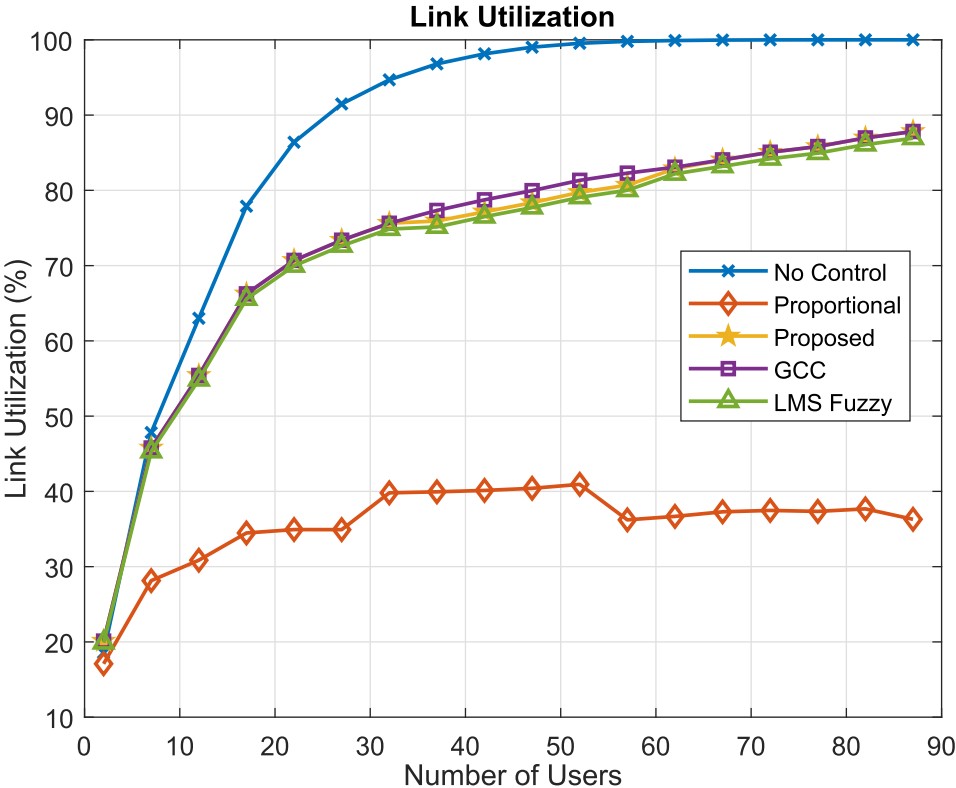

**Fig 11. Link utilization (%).**

complexity with the addition of $P$ orthonormal base functions once it is given by $O(nM + P^2)$. Fortunately, it was not necessary to use high values for $P$ to get interesting results. As future works, we intend to apply lower computational complexity algorithms to obtain the orthonormal basis function poles using the values of the autocorrelation function.

## Acknowledgments

This study was financed in part by the Coordenação de Aperfeiçoamento de Pessoal de Nível Superior—Brazil (CAPES)—Finance Code 001.

## Author Contributions

**Writing – original draft:** Alisson Assis Cardoso, Flávio Henrique Teles Vieira.

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
