## [Decision Letter · Decision Letter 0]

29 Aug 2019

PONE-D-19-20670

Adaptive Fuzzy Flow Rate Control Considering Multifractal Traffic Modeling and 5G Communications

PLOS ONE

Dear Mr Assis Cardoso,

Thank you for submitting your manuscript to PLOS ONE. After careful consideration, we feel that it has merit but does not fully meet PLOS ONE’s publication criteria as it currently stands. Therefore, we invite you to submit a revised version of the manuscript that addresses the points raised during the review process.

We would appreciate receiving your revised manuscript by Oct 13 2019 11:59PM. To enhance the reproducibility of your results, we recommend that if applicable you deposit your laboratory protocols in protocols.io, where a protocol can be assigned its own identifier (DOI) such that it can be cited independently in the future. For instructions see: http://journals.plos.org/plosone/s/submission-guidelines#loc-laboratory-protocols

We look forward to receiving your revised manuscript.

Kind regards,

Yong Wang

Academic Editor

PLOS ONE

Journal Requirements:

Reviewers' comments:

Reviewer's Responses to Questions

**Comments to the Author**

1. Is the manuscript technically sound, and do the data support the conclusions?

Reviewer #1: Yes

Reviewer #2: Yes

2. Has the statistical analysis been performed appropriately and rigorously? 

Reviewer #1: Yes

Reviewer #2: Yes

3. Have the authors made all data underlying the findings in their manuscript fully available?

Reviewer #1: Yes

Reviewer #2: No

4. Is the manuscript presented in an intelligible fashion and written in standard English?

Reviewer #1: Yes

Reviewer #2: Yes

5. Review Comments to the Author

Reviewer #1: This paper studied the multifractal traffic modeling using adaptive fuzzy flow rate control. Some conclusions and phenomena were found. This paper’s topic is interesting, but there are some aspects need to be improved from its current form: 1. The relevant studies in the literature review are not enough for current studies about the topic. 2. The problem description section should be added, and the main contribution should be further clearly presented. 3. The logic structure of this paper is not very unclear. 4. In the results and discussions section, I think that the authors should add more discussion to show the practical significance of the proposed research.

Therefore, I think this paper is interesting and the above aspects should be further improved.

Reviewer #2: This study introduced a OBF-Fuzzy flow control scheme for the 5G downlink. The topic of this study is interesting, the paper is written well and organized clearly. Before I address my acceptance of this study, several minor comments have to be pointed out.

(1) The introduction needs improvement. I suggest authors could summarize the limitation of current literatures, and make a correspondence with the contribution of this study.

(2) In the conclusion, limitations of this study need further discussion, and they may also give introduction for the future research directions.

(3) Several papers focusing on the network traffic modeling are still missing, I hope these following papers could be helpful for their future study.

[1] Short-term traffic flow prediction considering spatio-temporal correlation: a hybrid model combing type-2 fuzzy c-means and artificial neural network. IEEE Access, Vol. 7, No. 1, 101009-101018, 2019

[2]Traffic flow prediction based on combination of support vector machine and data denoising schemes. Physica A, 2019, Doi:10.1016/j.physa.2019.03.007

[3] Understanding characteristics in multivariate traffic flow time series from complex network structure. Physica A, Vol.477, 149-160, 2017

6. PLOS authors have the option to publish the peer review history of their article (what does this mean?). If published, this will include your full peer review and any attached files.

Reviewer #1: No

Reviewer #2: No

---

## [Author Response · Author response to Decision Letter 0]

17 Sep 2019

Dear Editor and Reviewers,

 We would like to thank you for the attention we received, for the comments and suggestions that have greatly contributed to the improvement of our work.

 As recommended, we carefully performed all suggested modifications and corrections as reported in response to reviewers.

 We are resubmitting this new corrected version, confident that we have fully attended the reviewers. We are at your complete disposal.

Best regards,

Alisson A. Cardoso and Flávio H. T. Vieira

Response to reviewers

Reviewer #1: This paper studied the multifractal traffic modeling using adaptive fuzzy flow rate control. Some conclusions and phenomena were found. This paper’s topic is interesting, but there are some aspects need to be improved from its current form: 1. The relevant studies in the literature review are not enough for current studies about the topic. 2. The problem description section should be added, and the main contribution should be further clearly presented. 3. The logic structure of this paper is not very unclear. 4. In the results and discussions section, I think that the authors should add more discussion to show the practical significance of the proposed research.

Response to the Reviewer #1: 

1) The article was improved by inserting into the introduction substantial content, such as, more descriptions of relevant related works. The following paragraphs were added to the introduction of the manuscript:

“Network traffic control can be enhanced when considering a precise traffic modeling, such as that provided by fuzzy approaches. Fuzzy modeling has been widely applied to many researches since it presents certain advantages over linear models, for example, in the description of unknown real processes with nonlinear and time-varying characteristics such as network traffic [3]. In [4], the authors propose an hybrid technique combining the Type-2 Fuzzy C-Means and Artificial Neural Network to improve the prediction of highway speed traffic flow compared to classical methods in the literature. In [5], it is proposed a prediction method that combines denoising schemes and support vector machine, outperforming models that do not consider the

denoising strategy. 

In the last decades, several studies have shown the importance of traffic process analysis using the wavelet transform due to its multiscale representation [5–8]. One of the applications of wavelet transform is in network traffic modeling in order to describe behaviors such as long-range dependence and burst incidences at different time scales [9, 10]. These characteristics may degrade network performance in relation to Gaussian and short-range dependence traffic flows [8, 11]. Multifractal models precisely describe traffic flows in small scales (ms or smaller), being adequate for the initial 5G systems, whose scheduling time is of the order of 1 ms [10, 12].

The main multifractal models are based on multiplicative cascades, which are structures where an interval is randomly divided by multipliers, conserving the interval mass [8]. Thus, at the end of the division process, a correlated sequence is obtained, representing the network traffic samples. As examples of wavelet domain based multifractal models, we can cite: The Lognormal Beta [13] model and the MWM (Multifractal Wavelet Model) [8]. The MWM model consists of a multiplicative cascade in the Haar wavelet domain [14], where multiplicative cascade multipliers are computed based on the signal energy decay. Although the MWM model being suitable for modeling network traffic, it requires the application of the wavelet transform to the whole traffic trace or to all samples in a time window that is intending to apply the model. In other words, in the original formulation of the MWM, its parameters are not updated at each time instant that a traffic sample is provided. This motivated us to propose an adaptive wavelet based multifractal modeling approach that is precise even being adequate for real time applications.

In order to achieve high utilization of resources in communication networks and for better decision making, traffic prediction can be used and must be as accurate as possible. Fuzzy modeling is capable of precisely representing a nonlinear complex process such as network traffic traces through the combination of linear local models [3]. In [15], the authors highlight the importance and principles of fuzzy logic applications in the area of channel estimation, channel equalization, handover management and QoS (Quality of Service) management. Moreover, adaptive prediction algorithms are more appropriate for real time multimedia applications than on-batch prediction algorithms due to on-line processing capability and varying nature of network traffic. Taking these into account, we also address the development of an adaptive fuzzy prediction algorithm that incorporates a wavelet domain modeling of network traffic.

In [16], the authors propose a scheduling algorithm with flow rate control for LTE downlink systems taking into account the size of each user queue. Thus, users with greater queue sizes will have higher priority compared to others. Also, in [16], the authors propose to use flow rate control algorithms to control network traffic that is not sensitive to delay (best effort). The results presented by the authors show that control algorithms can provide a significant improvement in the waiting time in the queues.

There are various proposals of control schemes in the literature that are dedicated to network protocols, such as that presented in [17], that is based on the flow control mechanisms of the Transmission Control Protocol/Internet Protocol (TCP/IP). Among the proposals for flow rate control that do not depend on specific network mechanisms, we can mention the Proportional Control method [17, 18]. Such methods can be used to control real-time applications and are also effective for other control problems.”

2) A section named “Problem Description” was created to better explain the main contribution of the paper.

3) A problem description section, a paragraph about the organization of the article and a paragraph of the main contributions of the paper have been added. The following paragraphs were added to the manuscript:

“This paper is divided as follows: In Section 2, we describe the problem of network flow rate control to improve QoS parameters, mainly regarding buffer occupation. We address the problem by considering fuzzy control techniques and multifractal modeling of traffic traces applied to 5G communications. Therefore, we first present in Section 3 a proposal of an algorithm to estimate the Lognormal Beta Model parameters in an adaptive manner. Next, in Section 4, concepts of orthonormal basis function, fuzzy logic and a proposal of flow rate control named GOBF-Fuzzy Flow Rate Control algorithm are presented. Regarding the wireless communication part of our work, in Section 5, we describe the 5G Downlink system (based on the first recommendations [1, 12]). In Section 6, we present the results obtained in the simulations with the considered traffic control algorithms. Finally, in Section 7, we conclude this work.”

and

“In summary, the main contributions of the present paper are:

1. Equations to adaptively estimate the moment factor and the scaling function of network traffic flows;

2. An algorithm possessing computer complexity O(1) to adaptively estimate the parameters of the Multifractal Lognormal Beta model;

3. A novel fuzzy flow rate control algorithm considering multifractal modeling and orthonormal basis functions that provides network performance improvement compared to others.”

4) In the results and discussions section, it was highlighted the advantages of the proposed flow control algorithm in the practical point of view. The following paragraphs were added to the manuscript:

“The flow control performance is an important factor for the communication quality of mobile networks. Incoming data can be stored in queues and if the buffer occupancy is increasing too much, data loss can occur and communication may be inefficient, requiring data retransmission and reducing the effective transmission data rate of users, which leads to lower baud rates than those stipulated in the contract with the operator. The results of the simulations carried out indicate that it is possible to obtain lower buffer occupancy values by using the proposed network traffic flow control algorithm compared to the others.”

and

“The delay in the network is an important QoS parameter that influences the number of users that can be allowed in the wireless system under the contract with the service provider. Let us assume, for example, a limiting value of 5ms. That is, data communication in the considered wireless network scenario should not exceed this value. It can be seen from Figure 8 that the proposed flow control algorithm can accommodate up to 87 users in the system, while the GCC control algorithm can allocate 52 users in the system; a difference of 35 users, that represents a significant financial loss for the mobile operator. By considering the case without flow control, it can be accommodated 22 users and 47 users using the proportional control algorithm and 82 users using the LMS Fuzzy algorithm, representing financial losses of 65, 40 users and 5, respectively.”

Reviewer #2: This study introduced a OBF-Fuzzy flow control scheme for the 5G downlink. The topic of this study is interesting, the paper is written well and organized clearly. Before I address my acceptance of this study, several minor comments have to be pointed out.

(1) The introduction needs improvement. I suggest authors could summarize the limitation of current literatures, and make a correspondence with the contribution of this study.

(2) In the conclusion, limitations of this study need further discussion, and they may also give introduction for the future research directions.

(3) Several papers focusing on the network traffic modeling are still missing, I hope these following papers could be helpful for their future study.

[1] Short-term traffic flow prediction considering spatio-temporal correlation: a hybrid model combing type-2 fuzzy c-means and artificial neural network. IEEE Access, Vol. 7, No. 1, 101009-101018, 2019

[2]Traffic flow prediction based on combination of support vector machine and data denoising schemes. Physica A, 2019, Doi:10.1016/j.physa.2019.03.007

[3] Understanding characteristics in multivariate traffic flow time series from complex network structure. Physica A, Vol.477, 149-160, 2017

Response to the Reviewer #2:

(1) The article was improved by inserting into the introduction substantial content, such as, more descriptions of relevant related works. The following paragraphs were added to the introduction of the manuscript:

“Network traffic control can be enhanced when considering a precise traffic modeling, such as that provided by fuzzy approaches. Fuzzy modeling has been widely applied to many researches since it presents certain advantages over linear models, for example, in the description of unknown real processes with nonlinear and time-varying characteristics such as network traffic [3]. In [4], the authors propose an hybrid technique combining the Type-2 Fuzzy C-Means and Artificial Neural Network to improve the prediction of highway speed traffic flow compared to classical methods in the literature. In [5], it is proposed a prediction method that combines denoising schemes and support vector machine, outperforming models that do not consider the

denoising strategy. 

In the last decades, several studies have shown the importance of traffic process analysis using the wavelet transform due to its multiscale representation [5–8]. One of the applications of wavelet transform is in network traffic modeling in order to describe behaviors such as long-range dependence and burst incidences at different time scales [9, 10]. These characteristics may degrade network performance in relation to Gaussian and short-range dependence traffic flows [8, 11]. Multifractal models precisely describe traffic flows in small scales (ms or smaller), being adequate for the initial 5G systems, whose scheduling time is of the order of 1 ms [10, 12].

The main multifractal models are based on multiplicative cascades, which are structures where an interval is randomly divided by multipliers, conserving the interval mass [8]. Thus, at the end of the division process, a correlated sequence is obtained, representing the network traffic samples. As examples of wavelet domain based multifractal models, we can cite: The Lognormal Beta [13] model and the MWM (Multifractal Wavelet Model) [8]. The MWM model consists of a multiplicative cascade in the Haar wavelet domain [14], where multiplicative cascade multipliers are computed based on the signal energy decay. Although the MWM model being suitable for modeling network traffic, it requires the application of the wavelet transform to the whole traffic trace or to all samples in a time window that is intending to apply the model. In other words, in the original formulation of the MWM, its parameters are not updated at each time instant that a traffic sample is provided. This motivated us to propose an adaptive wavelet based multifractal modeling approach that is precise even being adequate for real time applications.

In order to achieve high utilization of resources in communication networks and for better decision making, traffic prediction can be used and must be as accurate as possible. Fuzzy modeling is capable of precisely representing a nonlinear complex process such as network traffic traces through the combination of linear local models [3]. In [15], the authors highlight the importance and principles of fuzzy logic applications in the area of channel estimation, channel equalization, handover management and QoS (Quality of Service) management. Moreover, adaptive prediction algorithms are more appropriate for real time multimedia applications than on-batch prediction algorithms due to on-line processing capability and varying nature of network traffic. Taking these into account, we also address the development of an adaptive fuzzy prediction algorithm that incorporates a wavelet domain modeling of network traffic.

In [16], the authors propose a scheduling algorithm with flow rate control for LTE downlink systems taking into account the size of each user queue. Thus, users with greater queue sizes will have higher priority compared to others. Also, in [16], the authors propose to use flow rate control algorithms to control network traffic that is not sensitive to delay (best effort). The results presented by the authors show that control algorithms can provide a significant improvement in the waiting time in the queues.

There are various proposals of control schemes in the literature that are dedicated to network protocols, such as that presented in [17], that is based on the flow control mechanisms of the Transmission Control Protocol/Internet Protocol (TCP/IP). Among the proposals for flow rate control that do not depend on specific network mechanisms, we can mention the Proportional Control method [17, 18]. Such methods can be used to control real-time applications and are also effective for other control problems.”

(2) In the conclusion section, it was added and highlighted the limitation of the proposed flow control algorithm. Also, introduction for the future research directions was presented. The following paragraphs were added to the manuscript:

“The proposed algorithm provided better simulation results in terms of delay, buffer occupancy and loss rate than the other considered algorithms. The enhanced performance of the proposed fuzzy control algorithm is obtained at the cost of increasing the computational complexity with the addition of P orthonormal base functions once it is given by O(nM+P²). Fortunately, it was not necessary to use high values for P to get interesting results. As future works, we intend to apply lower computer complexity algorithms to obtain the orthonormal basis function poles using the values of the autocorrelation function.”

(3) We would like to thank the reviewers for their suggestions to the paper. We considered all the suggestions to improve the quality of the manuscript.

---

## [Decision Letter · Decision Letter 1]

24 Oct 2019

Adaptive Fuzzy Flow Rate Control Considering Multifractal Traffic Modeling and 5G Communications

PONE-D-19-20670R1

Dear Dr. Cardoso,

We are pleased to inform you that your manuscript has been judged scientifically suitable for publication and will be formally accepted for publication once it complies with all outstanding technical requirements.

With kind regards,

Yong Wang

Academic Editor

PLOS ONE

Additional Editor Comments (optional):

Reviewers' comments:

Reviewer's Responses to Questions

**Comments to the Author**

1. If the authors have adequately addressed your comments raised in a previous round of review and you feel that this manuscript is now acceptable for publication, you may indicate that here to bypass the “Comments to the Author” section, enter your conflict of interest statement in the “Confidential to Editor” section, and submit your "Accept" recommendation.

Reviewer #1: All comments have been addressed

Reviewer #2: All comments have been addressed

2. Is the manuscript technically sound, and do the data support the conclusions?

Reviewer #1: Yes

Reviewer #2: Yes

3. Has the statistical analysis been performed appropriately and rigorously? 

Reviewer #1: Yes

Reviewer #2: Yes

4. Have the authors made all data underlying the findings in their manuscript fully available?

Reviewer #1: Yes

Reviewer #2: No

5. Is the manuscript presented in an intelligible fashion and written in standard English?

Reviewer #1: Yes

Reviewer #2: Yes

6. Review Comments to the Author

Reviewer #1: The comments from the reviewer have been addressed in the revised manuscript. Now the paper looks good.

Reviewer #2: In this revision, authors responded all my concerning comments, and I think the quality of the paper has improved largely and it can be accepted for its current condition.

7. PLOS authors have the option to publish the peer review history of their article (what does this mean?). If published, this will include your full peer review and any attached files.

Reviewer #1: No

Reviewer #2: No

---

## [Editor Report · Acceptance letter]

29 Oct 2019

PONE-D-19-20670R1 

Adaptive Fuzzy Flow Rate Control Considering Multifractal Traffic Modeling and 5G Communications 

Dear Dr. Cardoso:

I am pleased to inform you that your manuscript has been deemed suitable for publication in PLOS ONE. Congratulations! Your manuscript is now with our production department. 

With kind regards,

on behalf of

Dr. Yong Wang 

Academic Editor

PLOS ONE